# Modeling Dynamic Functional Connectivity with Latent Factor Gaussian Processes

**Lingge Li**[*]
UC Irvine
linggel@uci.edu

**Dustin Pluta**[*]
UC Irvine
dpluta@uci.edu

**Babak Shahbaba**
UC Irvine
babaks@uci.edu

**Norbert Fortin**
UC Irvine
norbert.fortin@uci.edu

**Hernando Ombao**
KAUST
hernando.ombao@kaust.edu.sa

**Pierre Baldi**
UC Irvine
pfbaldi@ics.uci.edu

## Abstract

Dynamic functional connectivity, as measured by the time-varying covariance of neurological signals, is believed to play an important role in many aspects of cognition. While many methods have been proposed, reliably establishing the presence and characteristics of brain connectivity is challenging due to the high dimensionality and noisiness of neuroimaging data. We present a latent factor Gaussian process model which addresses these challenges by learning a parsimonious representation of connectivity dynamics. The proposed model naturally allows for inference and visualization of connectivity dynamics. As an illustration of the scientific utility of the model, application to a data set of rat local field potential activity recorded during a complex non-spatial memory task provides evidence of stimuli differentiation.

## 1 Introduction

The celebrated discoveries of place cells, grid cells, and similar structures in the hippocampus have produced a detailed, experimentally validated theory of the formation and processing of spatial memories. However, the specific characteristics of non-spatial memories, e.g. memories of odors and sounds, are still poorly understood. Recent results from human fMRI and EEG experiments suggest that dynamic functional connectivity (DFC) is important for the encoding and retrieval of memories [1, 2, 3, 4, 5, 6], yet DFC in local field potentials (LFP) in animal models has received relatively little attention. We here propose a novel latent factor Gaussian process (LFGP) model for DFC estimation and apply it to a data set of rat hippocampus LFP during a non-spatial memory task [7]. The model produces strong statistical evidence for DFC and finds distinctive patterns of DFC associated with different experimental stimuli.

Due to the high-dimensionality of time-varying covariance and the complex nature of cognitive processes, effective analysis of DFC requires balancing model parsimony, flexibility, and robustness to noise. DFC models fall into a common framework with three key elements: dimensionality reduction, covariance estimation from time series, and identification of connectivity patterns [8]. Many neuroimaging studies use a combination of various methods, such as sliding window (SW) estimation, principal component analysis (PCA), and the hidden Markov model (HMM) (see e.g. [9, 10, 11]). In general, these methods are not fully probabilistic, which can make uncertainty quantification and inference difficult in practice.

[*]These authors contributed equally to this work.

Bayesian latent factor models provide a probabilistic approach to modeling dynamic covariance that allows for simultaneous dimensionality reduction and covariance process estimation. Examples include the latent factor stochastic volatility (LFSV) model [12] and the nonparametric covariance model [13]. In the LFSV model, an autoregressive process is imposed on the latent factors and can be overly restrictive. While the nonparametric model is considerably more flexible, the matrix process for time-varying loadings adds substantial complexity.

Aiming to bridge the gap between these factor models, we propose the latent factor Gaussian process (LFGP) model. In this approach, a latent factor structure is placed on the log-covariance process of a non-stationary multivariate time series, rather than on the observed time series itself as in other factor models. Since covariance matrices lie on the manifold of symmetric positive-definite (SPD) matrices, we utilize the Log-Euclidean metric to allow unconstrained modeling of the vectorized upper triangle of the covariance process. Dimension reduction and model parsimony is achieved by representing each covariance element as a linear combination of Gaussian process latent factors [14].

In this work, we highlight three major advantages of the LFGP model for practical DFC analysis. First, through the prior on the Gaussian process length scale, we are able to incorporate scientific knowledge to target specific frequency ranges that are of scientific interest. Second, the model posterior allows us to perform Bayesian inference for scientific hypotheses, for instance, whether the LFP time series is non-stationary, and if characteristics of DFC differ across experimental conditions. Third, the latent factors serve as a low-dimensional representation of the covariance process, which facilitates visualization of complex phenomena of scientific interest, such as the role of DFC in stimuli discrimination in the context of a non-spatial memory experiment.

## 2 Background

### 2.1 Sliding Window Covariance Estimation

Sliding window methods have been extensively researched for the estimation and analysis of DFC, particularly in human fMRI studies; applications of these methods have identified significant associations of DFC with disease status, behavioral outcomes, and cognitive differences in humans. See [8] for a recent detailed review of existing literature. For $X(t) \sim \mathcal{N}(0, K(t))$ a $p$-variate time series of length $T$ with covariance process $K(t)$, the sliding window covariance estimate $\hat{K}_{SW}(t)$ with window length $L$ can be written as the convolution $\hat{K}_{SW}(t) = (h * XX')(t) = \sum_{s=1}^{T} h(s)X(t-s)X(t-s)' \, ds$, for the rectangular kernel $h(t) = \mathbf{1}_{[0,L-1]}(t)/L$, where $\mathbf{1}$ is the indicator function. Studies of the performance of sliding window estimates recommend the use of a tapered kernel to decrease the impact of outlying measurements and to improve the spectral properties of the estimate [15, 16, 17]. In the present work we employ a Gaussian taper with scale $\tau$ defined as $h^\tau(t) = \frac{1}{\zeta} \exp\left\{-\frac{1}{2}\left(\frac{t-L/2}{\tau L/2}\right)^2\right\} \mathbf{1}_{[0,L-1]}(t)$, where $\zeta$ is a normalizing constant. The corresponding tapered SW estimate is $\hat{K}_\tau(t) = (h^\tau * XX')(t)$.

### 2.2 Log-Euclidean Metric

Direct modeling of the covariance process from the SW estimates is complicated by the positive definite constraint of the covariance matrices. To ensure the model estimates are positive definite, it is necessary to employ post-hoc adjustments, or to build the constraints into the model, typically by utilizing the Cholesky or spectral decompositions. The LFGP model instead uses the Log-Euclidean framework of symmetric positive definite (SPD) matrices to naturally ensure positive-definiteness of the estimated covariance process while also simplifying the model formulation and implementation.

Denote the space of $p \times p$ SPD matrices as $\mathbb{P}_p$. For $X_1, X_2 \in \mathbb{P}_p$, the *Log-Euclidean* distance is defined by $d_{LE}(X_1, X_2) = \|\text{Log}(X_1) - \text{Log}(X_2)\|$, where Log is the matrix logarithm, and $\|\cdot\|$ is the Frobenius norm. The metric space $(\mathbb{P}_p, d_{LE})$ is a Riemannian manifold that is isomorphic to $\mathbb{R}^q$ with the usual Euclidean norm, for $q = (p+1)p/2$.

Methods for modeling covariances in regression contexts via the matrix logarithm were first introduced in [18]. The Log-Euclidean framework for analysis of SPD matrices in neuroimaging contexts was first proposed in [19], with further applications in neuroimaging having been developed in recent years [20]. The present work is a novel application of the Log-Euclidean framework for DFC analysis.

## 2.3 Bayesian Latent Factor Models

For $x_{ij}, i = 1, \ldots, n, j = 1, \ldots, p$, the simple Bayesian latent factor model is $x_i = f_i \Lambda + \varepsilon_i$, with $f_i \overset{iid}{\sim} \mathcal{N}(0, I_r), \varepsilon_i \overset{iid}{\sim} \mathcal{N}(0, \Sigma)$, and $\Lambda$ an $r \times p$ matrix of factor loadings [21]. $\Sigma$ is commonly assumed to be a diagonal matrix, implying the latent factors capture all the correlation structure of the $p$ features of $x$. The latent factor model shares some similarities with principal component analysis, but includes a stochastic error term, which leads to a different interpretation of the resulting factors [9, 10].

Variants of the linear factor model have been developed for modeling non-stationary multivariate time series [22, 23]. In general, these models represent the $p$-variate observed time series as a linear combination of $r$ latent factors $f_j(t), j = 1, \ldots, r$, with $r \times q$ loading matrix $\Lambda$ and errors $\varepsilon(t)$: $X(t) = f(t)\Lambda + \varepsilon(t)$. From this general modeling framework, numerous methods for capturing the non-stationary dynamics in the underlying time series have been developed, such as latent factor stochastic volatility (LFSV) [12], dynamic conditional correlation [24], and the nonparametric covariance model [13].

## 2.4 Gaussian Processes

A Gaussian process ($\mathcal{GP}$) is a continuous stochastic process for which any finite collection of points are jointly Gaussian with some specified mean and covariance. A $\mathcal{GP}$ can be understood as a distribution on functions belonging to a particular reproducing kernel Hilbert space (RKHS) determined by the covariance operator of the process [25]. Typically, a zero mean $\mathcal{GP}$ is assumed (i.e. the functional data has been centered by subtracting a consistent estimator of the mean), so that the $\mathcal{GP}$ is parameterized entirely by the kernel function $\kappa$ that defines the pairwise covariance. Let $f \sim \mathcal{GP}(0, k(\cdot, \cdot))$. Then for any $x$ and $x'$ we have

$$\left( \begin{array}{c} f(x) \\ f(x') \end{array} \right) \sim \mathcal{N} \left( 0, \left[ \begin{array}{cc} \kappa(x, x) & \kappa(x, x') \\ \kappa(x, x') & \kappa(x', x') \end{array} \right] \right). \tag{1}$$

Further details are given in [26].

# 3 Latent Factor Gaussian Process Model

## 3.1 Formulation

We consider estimation of dynamic covariance from a sample of $n$ independent time series with $p$ variables and $T$ time points. Denote the $i$th observed $p$-variate time series by $X_i(t), i = 1, \cdots, n$. We assume that each $X_i(t)$ follows an independent distribution $\mathcal{D}$ with zero mean and stochastic covariance process $K_i(t)$. To model the covariance process, we first compute the Gaussian tapered sliding window covariance estimates for each $X_i(t)$, with fixed window size $L$ and taper $\tau$ to obtain $\hat{K}_{\tau,i}$. We then apply the matrix logarithm to obtain the $q = p(p+1)/2$ length vector $Y_i(t)$ specified by $\hat{K}_{\tau,i} = \text{Log}(\vec{\mathbf{u}}(Y_i))$, where $\vec{\mathbf{u}}$ maps a matrix to its vectorized upper triangle. We refer to $Y_i(t)$ as the "log-covariance" at time $t$.

The resulting $Y_i(t)$ can be modeled as an unconstrained $q$-variate time series. The LFGP model represents $Y_i(t)$ as a linear combination of $r$ latent factors $F_i(t)$ through an $r \times q$ loading matrix $B$ and independent Gaussian errors $\epsilon_i$. The loading matrix $B$ is held constant across observations and time. Here $F_i(t)$ is modeled as a product of independent Gaussian processes. Placing priors $p_1, p_2, p_3$ on the loading matrix $B$, Gaussian noise variance $\sigma^2$, and Gaussian process hyper-parameter $\theta$, respectively, gives a fully probabilistic latent factor model on the covariance process:

$$X_i(t) \sim \mathcal{D}(0, K_i(t)) \text{ where } K_i(t) = \exp\left(\vec{\mathbf{u}}(Y_i(t))\right) \tag{2}$$

$$Y_i(t) = F_i(t) \cdot B + \epsilon_i \text{ where } \epsilon_i \overset{iid}{\sim} \mathcal{N}(0, I\sigma^2) \tag{3}$$

$$F_i(t) \sim \mathcal{GP}(0, \kappa(t; \theta)) \tag{4}$$

$$B \sim p_1, \sigma^2 \sim p_2, \theta \sim p_3. \tag{5}$$

The LFGP model employs a latent distribution of curves $\mathcal{GP}(0, \kappa(t; \theta))$ to capture temporal dependence of the covariance process, thus inducing a Gaussian process on the log-covariance $Y(t)$. This

conveniently allows multiple observations to be modeled as different realizations of the same induced $\mathcal{GP}$ as done in [27]. The model posteriors are conditioned on different observations despite sharing the same kernel. For better identifiability, the $\mathcal{GP}$ variance scale is fixed so that the loading matrix can be unconstrained.

## 3.2 Properties

**Theorem 1.** *The log-covariance process induced by the LFGP model is weakly stationary when the GP kernel $\kappa(s,t)$ depends only on $|s-t|$.*

*Proof.* The covariance of the log-covariance process $Y(t)$ depends only on the static loading matrix $B = (\beta_{kj})_{1 \le k \le r; 1 \le j \le q}$ and the factor covariance kernels. Explicitly, for factor kernels $\kappa(s,t;\theta_k), k = 1, \ldots, r$, and assuming $\varepsilon_i(t) \overset{iid}{\sim} \mathcal{N}(0, \Sigma)$, with $\Sigma = (\sigma_{jj'}^2)_{j,j' \le q}$ constant across observations and time, the covariance of elements of $Y(t)$ is

$$\text{Cov}(Y_{ij}(s), Y_{ij'}(t)) = \text{Cov}\left(\sum_{k=1}^{r} F_{ik}(s)\beta_{kj} + \varepsilon_{ij'}(t), \sum_{k=1}^{r} F_{ik}(t)\beta_{kj'} + \varepsilon_{ij'}(t)\right) \qquad (6)$$

$$= \sum_{k=1}^{r} \beta_{kj}\beta_{kj'}\kappa(s,t;\theta_k) + \sigma_{jj'}^2, \qquad (7)$$

which is weakly stationary when $\kappa(s,t)$ depends only on $|s-t|$. $\qquad\square$

***Posterior contraction.*** To consider posterior contraction of the LFGP model, we make the following assumptions. The true log-covariance process $w = \vec{\mathbf{u}}(\log(K(t)))$ is in the support of the product $\mathcal{GP}$ $W \sim F(t)B$, for $F(t)$ and $B$ defined above, with known number of latent factors $r$. The $\mathcal{GP}$ kernel $\kappa$ is $\alpha$-Hölder continuous with $\alpha \ge 1/2$. $Y(t) : [0,1] \to \mathbb{R}^q$ is a smooth function in $\ell_q^\infty([0,1])$ with respect to the Euclidean norm, and the prior $p_2$ for $\sigma^2$ has support on a given interval $[a,b] \subset (0,\infty)$. Under the above assumptions, bounds on the posterior contraction rates then follow from previous results on posterior contraction of Gaussian process regression for $\alpha$-smooth functions given in [28, 29]. Specifically,

$$E_0\Pi_n((w,\sigma) : \|w - w_0\|_n + |\sigma - \sigma_0| > M\varepsilon_n|Y_1, \cdots, Y_n) \to 0$$

for sufficiently large $M$ and with posterior contraction rate $\varepsilon_n = n^{-\alpha/(2\alpha+q)} \log^\delta(n)$ for some $\delta > 0$, where $E_0(\Pi_n(\cdot|Y_1, \cdots, Y_n))$ is the expectation of the posterior under the model priors.

To illustrate posterior contraction in the LFGP model, we simulate data for five signals with various sample sizes ($n$) and numbers of observation time points ($t$), with a covariance process generated by two latent factors. To measure model bias, we consider the mean squared error of posterior median of the reconstructed log-covariance series. To measure posterior uncertainty, the posterior sample variance is used. As shown in Table 1, both sample size $n$ and number of observation time points $t$ contribute to posterior contraction.

Table 1: Mean squared error of posterior median (posterior sample variance) $\times 10^{-2}$

|  | $n = 1$ | $n = 10$ | $n = 20$ | $n = 50$ |
|---|---|---|---|---|
| $t = 25$ | 12.212 (20.225) | 7.845 (8.743) | 7.089 (7.714) | 5.869 (7.358) |
| $t = 50$ | 6.911 (7.588) | 4.123 (5.836) | 3.273 (3.989) | 3.237 (3.709) |
| $t = 100$ | 3.728 (5.218) | 1.682 (2.582) | 1.672 (2.659) | 1.672 (1.907) |

***Large prior support.*** The prior distribution of the log-covariance process $Y(t)$ is a linear combination of $r$ independent $\mathcal{GP}$s each with mean 0 and kernel $\kappa(s,t;\theta_k), k = 1, \cdots, r$. That is, each log-covariance element will have prior $Y_j(t) = \sum_{k=1}^{r} \beta_{jk}F_k(t) \sim \mathcal{GP}(0, \sum \beta_{jk}^2\kappa(s,t;\theta_k))$. Considering $B$ fixed, the resulting prior for $F_i(t)B$ has support equal to the closure of the reproducing kernel Hilbert space (RKHS) with kernel $B^T\mathcal{K}(t,\cdot)B$ [26], where $\mathcal{K}$ is the covariance tensor formed by stacking $\kappa_k = \kappa(s,t;\theta_k), k = 1, \cdots, r$ [25]. Accounting for the prior $p_1$ of $B$, a function

$W \in \ell_q^\infty[0,1]$ will have nonzero prior probability $\Pi_0(W) > 0$ if $W$ is in the closure of the RKHS with kernel $A^T \mathcal{K}(t, \cdot) A$ for some $A$ in the support of $p_1$.

## 3.3 Factor Selection via the Horseshoe Prior

Similar to other factor models, the number of latent factors in the LFGP model has a crucial effect on model performance, and must be selected somehow. For Bayesian factor analysis, there is extensive literature on factor selection methods, such as Bayes factors, reversible jump sampling [30], and shrinkage priors [31]. While we can compare different models in terms of goodness-of-fit, we cannot compare their latent factors in a meaningful way due to identifiability issues. Therefore, we instead iteratively increase the number of factors and fit the new factors on the residuals resulting from the previous fit. In order to avoid overfitting with too many factors, we place a horseshoe prior on the loadings of the new factors, so that the loadings shrink to zero if the new factor is unnecessary.

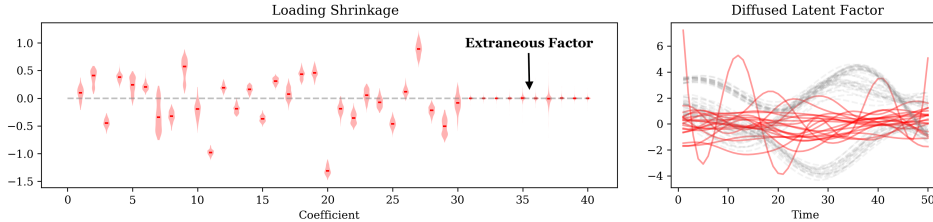

Figure 1: Violin plots of loading posteriors show that the loadings for the fourth factor (indices 30 to 39) shrink to zero with the horseshoe prior (left). Compared to the posteriors of the first three factors (dashed gray), the posterior of the extraneous factor (solid red) is diffused around zero as a result of zero loadings (right).

Introduced by [32], the horseshoe prior in the regression setting is given by

$$\beta | \lambda, \tau \sim N(0, \lambda^2 \rho^2) \tag{8}$$

$$\lambda \sim Cauchy^+(0, 1) \tag{9}$$

and can be considered as a scale-mixture of Gaussian distributions. A small global scale $\rho$ encourages shrinkage, while the heavy tailed Cauchy distribution allows the loadings to escape from zero. The example shown in Figure 1 illustrates the shrinkage effect of the horseshoe prior when iteratively fitting an LFGP model with four factors to simulated data generated from three latent factors. For sampling from the loading posterior distribution, we use the No-U-Turn Sampler [33] as implemented in PyStan [34].

## 3.4 Scalable Computation

The LFGP model can be fit via Gibbs sampling, as commonly done for Bayesian latent variable models. In every iteration, we first sample $F | B, \sigma^2, \theta, Y$ from the conditional $p(F|Y)$ as $F, Y$ are jointly multivariate Gaussian where the covariance can be written in terms of $B, \sigma^2, \theta$. However, it is worth noting that this multivariate Gaussian has a large covariance matrix, which could be computationally expensive to invert. Given $F$, the parameters $B, \sigma^2$ and $\theta$ become conditionally independent. Using conjugate priors for Bayesian linear regression, the posterior $p(B, \sigma^2|F, Y)$ is directly available. For the $\mathcal{GP}$ parameter posterior $p(\theta|F)$, either Metropolis random walk or slice sampling [35] can be used within each Gibbs step because the parameter space is low dimensional.

For efficient $\mathcal{GP}$ posterior sampling, it is essential to exploit the structure of the covariance matrix. For each independent latent $\mathcal{GP}$ factor $F_j$, there are $n$ independent sets of observations at $t$ time points. Therefore, the $\mathcal{GP}$ covariance matrix $\Sigma_j$ has dimensions $nT \times nT$. To reduce the computational burden, we notice that the covariance $\Sigma_j$ can be decomposed using a Kronecker product $\Sigma_j = I_n \otimes K_{time}(t)$, where $K_{time}$ is the $T \times T$ temporal covariance. The cost to invert $\Sigma_j$ using this decomposition is $\mathcal{O}(T^3)$, which is a substantial reduction compared to the original cost $\mathcal{O}((nT)^3)$. For many choices of kernel, such as the squared-exponential or Matérn kernel, $K_{time}(t)$ has a Toeplitz structure and can be approximated through interpolation [36], further reducing the computational cost.

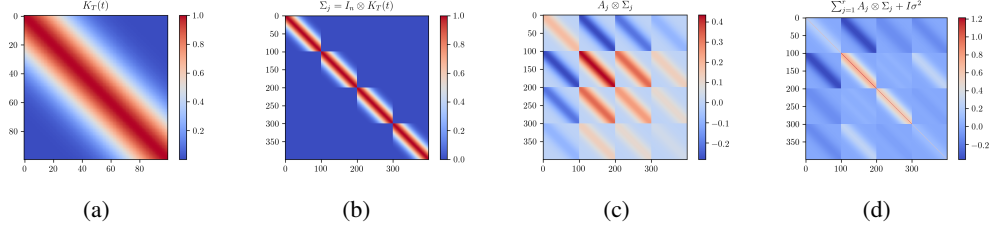

|     |     |     |     |
|:---:|:---:|:---:|:---:|
| (a) | (b) | (c) | (d) |

Figure 2: The full covariance matrix $\Sigma_Y$ is composed of building blocks of smaller matrices. (a) $\mathcal{GP}$ covariance matrix at evenly-spaced time points, (b) covariance matrix of factor $F_j$ for $n$ sets of observations, (c) contribution to the covariance of $Y$ from factor $F_j$, and (d) full covariance matrix $\Sigma_Y$.

Combining the latent $\mathcal{GP}$ factors $F$ (dimensions $n \times T \times r$) and loading matrix $B$ (dimensions $r \times q$) induces a $\mathcal{GP}$ on $Y$. The dimensionality of $Y$ is $n \times T \times q$ so the full $(nTq) \times (nTq)$ covariance matrix $\Sigma_Y$ is prohibitive to invert. As every column of Y is a weighted sum of the $\mathcal{GP}$ factors, the covariance matrix $\Sigma_Y$ can be written as a sum of Kronecker products $\sum_{j=1}^{r} A_j \otimes \Sigma_j + I\sigma^2$, where $\Sigma_j$ is the covariance matrix of the $j$th latent $\mathcal{GP}$ factor and $A_j$ is a $q \times q$ matrix based on the factor loadings. We can regress residuals of $Y$ on each column of $F$ iteratively to sample from the conditional distribution $p(F|Y)$ so that the residual covariance is only $A_j \otimes \Sigma_j + I$. The inversion can be done in a computationally efficient way with the following matrix identity

$$(C \otimes D + I)^{-1} = (P \otimes Q)^T (I + \Lambda_1 \otimes \Lambda_2)^{-1} (P \otimes Q) \tag{10}$$

where $C = P\Lambda_1 P^T$ and $D = Q\Lambda_2 Q^T$ are the spectral decompositions. In the identity, obtaining $P, Q, \Lambda_1, \Lambda_2$ costs $\mathcal{O}(q^3)$ and $\mathcal{O}((nT)^3)$, which is a substantial reduction from the cost of direct inversion, $\mathcal{O}((nTq)^3)$; calculating $(I + \Lambda_1 \otimes \Lambda_2)^{-1}$ is straightforward since $\Lambda_1$ and $\Lambda_2$ are diagonal.

## 4    Experiments

### 4.1    Model Comparisons on Simulated Data

We here consider three benchmark models: sliding window with principal component analysis (SW-PCA), hidden Markov model, and LFSV model. SW-PCA and HMM are commonly used in DFC studies but have severe limitations. The sliding window covariance estimates are consistent but noisy, and PCA does not take the estimation error into account. HMM is a probabilistic model and can be used in conjunction with a time series model, but it is not well-suited to capturing smoothly varying dynamics in brain connectivity.

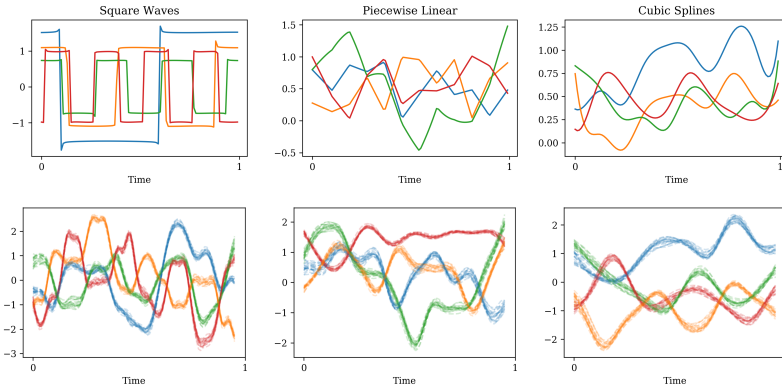

Figure 3: With the jagged dynamics of discrete states, the LFGP model fails to capture the "jumps" but approximates the overall trend (left). When the underlying dynamics are smooth, the LFGP model can accurately recover the shape up to some scaling constant (right).

To compare the performance of different models, we simulate time series data $X_t \sim N(0, K(t))$ with time-varying covariance $K(t)$. The covariance $K(t)$ follows deterministic dynamics that are given by $\vec{\mathbf{u}}(\log(K(t))) = U(t) \cdot A$. We consider three different scenarios of dynamics $U(t)$: square waves, piece-wise linear functions, and cubic splines. Note that both square waves and piece-wise linear functions give rise to dynamics that are not well-represented by the LFGP model when the squared-exponential kernel is used. For each scenario, we randomly generate 100 time series data sets and fit all the models. The evaluation metric is reconstruction loss of the covariance as measured by the Log-Euclidean distance. The simulation results in Table 2 show that the proposed LFGP model has the lowest reconstruction loss among the methods considered. Each time series has 10 variables

Table 2: Median reconstruction loss (standard deviation) across 100 data sets

|  | SW-PCA | HMM | LFSV | LFGP |
|---|---|---|---|---|
| Square save | 0.693 (0.499) | 1.003 (1.299) | 4.458 (2.416) | 0.380 (0.420) |
| Piece-wise | 0.034 (0.093) | 0.130 (0.124) | 0.660 (0.890) | 0.027 (0.088) |
| Smooth spline | 0.037 (0.016) | 0.137 (0.113) | 0.532 (0.400) | 0.028 (0.123) |

with 1000 observations and the latent dynamics are 4-dimensional as illustrated in Figure 3. For the SW-PCA model, the sliding window size is 50 and the number of principal components is 4. For the HMM, the number of hidden states is increased gradually until the model does not converge, following the implementation outlined in [37]. For the LFSV model, the R package *factorstochvol* is used with default settings. All simulations are run on a 2.7 GHz Intel Core i5 Macbook Pro laptop with 8GB memory.

## 4.2 Application to Rat Hippocampus Local Field Potentials

To investigate the neural mechanisms underlying the temporal organization of memories, [7] recorded neural activity in the CA1 region of the hippocampus as rats performed a sequence memory task. The task involves the presentation of repeated sequences of 5 stimuli (odors A, B, C, D, and E) at a single port and requires animals to correctly identify each stimulus as being presented either "in sequence" (e.g., ABC...) or "out of sequence" (e.g., ABD...) to receive a reward. Here the model is applied to local field potential (LFP) activity recorded from the rat hippocampus, but the key reason for choosing this data set is that it provides a rare opportunity to subsequently apply the model to other forms of neural activity data collected using the same task (including spiking activity from different regions in rats [38] and whole-brain fMRI in humans).

LFP signals were recorded in the hippocampi of five rats performing the task. The local field potentials are measured by surgically implanted tetrodes and the exact tetrode locations vary across rats. Therefore, it may not make sense to compare LFP channels of different rats. This issue actually motivates the latent factor approach because we want to eventually visualize and compare the latent trajectories for all the rats. For the present analysis, we have focused on the data from a particular rat exhibiting the best memory task performance. To boost the signal-to-noise ratio, six LFP channels that recorded a majority of the attached neurons were chosen. Only trials of odors B and C were considered, to avoid potential confounders with odor A being the first odor presented, and due to substantially fewer trials for odors D and E.

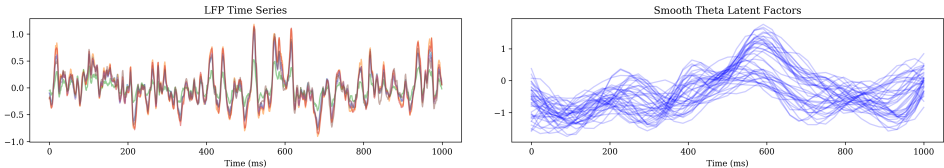

Figure 4: Time series of 6 LFP channels for a single trial sampled at 1000Hz include all frequency components (left). Posterior draws of latent factors for the covariance process appear to be smoothly varying near the theta frequency range (right).

During each trial, the LFP signals are sampled at 1000Hz for one second after odor release. We focus on 41 trials of odor B and 37 trials of odor C. Figure 4 shows the time series of these six LFP channels

for a single trial. We treat all 78 trials as different realizations of the same stochastic process without distinguishing the stimuli explicitly in the model. In order to facilitate interpretation of the latent space representation, we fit two latent factors which explain about 40% of the variance in the data. The prior for $\mathcal{GP}$ length scale is a Gamma distribution concentrated around 100ms on the time scale to encourage learning frequency dynamics close to the theta range (4-12 Hz). Notably, oscillations in this frequency range have been associated with memory function but have not previously been shown to differentiate among the type of stimuli used here, thus providing an opportunity to test the sensitivity of the model. For the loadings and variances, we use the Gaussian-Inverse Gamma conjugate priors. 20,000 MCMC draws are taken with the first 5000 draws discarded as burn-in.

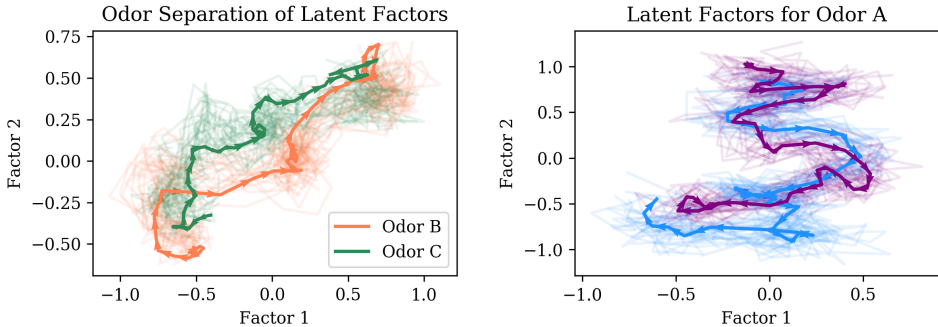

Figure 5: Posterior draws of median $\mathcal{GP}$ factors visualized as trajectories in latent space can be separated based on the odor, with maximum separation around 250ms (left). The latent trajectories are much more intertwined when the model is fitted to data of the same odor. (right)

For each odor, we can calculate the posterior median latent factors across trials and visualize them as a trajectory in the latent space. Figure 5 shows that the two trajectories start in an almost overlapping area, with separation occurring around 250ms. This is corroborated by the experimental data indicating that animals begin to identify the odor 200-250ms after onset. We also observe that the two trajectories converge toward the end of the odor presentation. This is also consistent with the experimental data showing that, by then, animals have correctly identified the odors and are simply waiting to perform the response (thereby resulting in similar neural states). In order to quantify odor separation, we can evaluate the difference between the posterior distributions of odor median latent trajectories by using classifiers on the MCMC draws. We also fit the model to two random subsets of the 58 trials of odor A and train the same classifiers. Table 3) shows the classification results and the posteriors are more separated for different odors.

Table 3: Odor separation as measured by Latent space classification accuracy (standard deviation)

|  | Different odors | Same odor |
| --- | --- | --- |
| Logistic regression | 69.97 (0.78) | 63.10 (0.91) |
| k-NN | 87.12 (0.33) | 78.41 (0.65) |
| SVM | 74.53 (0.67) | 64.75 (1.21) |

As a comparison, a hidden Markov model was fit to the LFP data from the same six selected tetrodes. Figure 6 compares the estimated covariance with different models. Eight states were selected with an elbow method using the AIC of the HMM; we note that the minimum AIC is not achieved for less than 50 states, suggesting that the dynamics of the LFP covariance may be better described with a continuous model. Moreover, the proportion of time spent in each state for odor B and C trials given in Table 4 fails to capture odor separation in the LFP data.

Collectively, these results provide compelling evidence that this model can use LFP activity to differentiate the representation of different stimuli, as well as capture their expected dynamics within trials. Stimuli differentiation has frequently been accomplished by analyzing spiking activity, but not LFP activity alone. This approach, which may be applicable to other types of neural data including spiking activity and fMRI activity, may significantly advance our ability to understand how information is represented among brain regions.

Table 4: State proportions for odors B and C as estimated by HMM

| Odor | State 1 | State 2 | State 3 | State 4 | State 5 | State 6 | State 7 | State 8 |
|------|---------|---------|---------|---------|---------|---------|---------|---------|
| B | 0.123 | 0.089 | 0.146 | 0.153 | 0.109 | 0.159 | 0.160 | 0.061 |
| C | 0.133 | 0.092 | 0.144 | 0.147 | 0.106 | 0.164 | 0.152 | 0.062 |

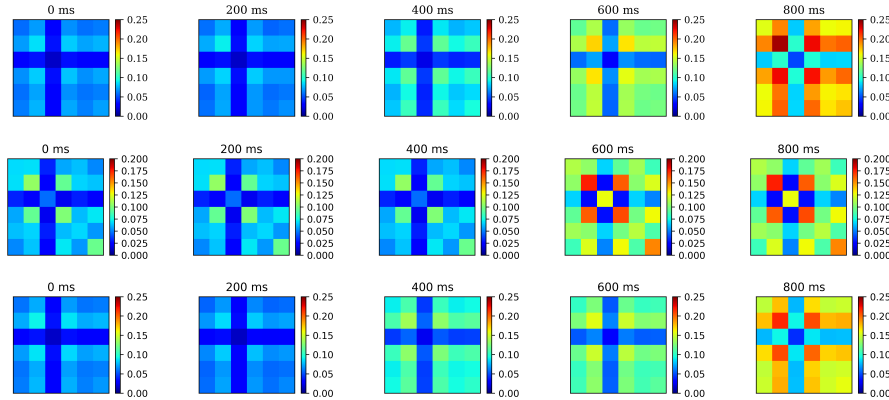

Figure 6: Median covariance matrices over time for odor B trials estimated with sliding window (top), HMM (middle), and LFGP model (bottom) reveal similar patterns in dynamic connectivity in the six LFP channels.

## 5 Discussion

The proposed LFGP model is a novel application of latent factor models for directly modeling the dynamic covariance in multivariate non-stationary time series. As a fully probabilistic approach, the model naturally allows for inference regarding the presence of DFC, and for detecting differences in connectivity across experimental conditions. Moreover, the latent factor structure enables visualization and scientific interpretation of connectivity patterns. Currently, the main limitation of the model is scalability with respect to the number of observed signals. Thus, in practical applications it may be necessary to select a relevant subset of the observed signals, or apply some form of clustering of similar signals. Future work will consider simultaneously reducing the dimension of the signals and modeling the covariance process to improve the scalability and performance of the LFGP model.

The Gaussian process regression framework is a new avenue for analysis of DFC in many neuroimaging modalities. Within this framework, it is possible to incorporate other covariates in the kernel function to naturally account for between-subject variability. In our setting, multiple trials are treated as independent observations or repeated measurements from the same rat, while in human neuroimaging studies, there are often single observations from many subjects. Pooling information across subjects in this setting could yield more efficient inference and lead to more generalizable results.

**Acknowledgments**

This work was supported by NIH award R01-MH115697 (B.S., H.O., N.J.F), NSF award DMS-1622490 (B.S.), Whitehall Foundation Award 2010-05-84 (N.J.F.), NSF CAREER award IOS-1150292 (N.J.F.), NSF award BSC-1439267 (N.J.F.), and KAUST research fund (H.O.). We would like to thank Michele Guindani (UC-Irvine), Weining Shen (UC-Irvine), and Moo Chung (Univ. of Wisconsin) for their helpful comments regarding this work.

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
