[Reviews · NeurIPS 2019]

Reviewer 1



This paper devises a new approach to dynamic functional connectivity modeling in a Gaussian process framework. One major novelty of the work is to formulate in a fully Bayesian framework by exploiting log matrix representation, using latent factor representation, and kernel representation jointly. There are several issues that should be clarified: - It is not clear how the loading factor matrix B is estimated. What type of prior distribution was used? Did the authors use conjugate distributions? - There are many hyperparameters in the model. But there is no information about those values used in the experiments. - Why did the authors focus on only trials of odor B and C, while there were 5 different odors in the data set? - The learned or estimated loading factor matrix B is not presented and analyzed at all.

Reviewer 2



Update: I appreciate the authors' response. Besides other reviewers' concerns, the current response is still not very substantial to me, especially on the experimental results. E.g., in the response, the authors claim that "when they fit the model to two random subsets of the 58 trials of the same odor A, the latent trajectories are much more intertwined". Comparing with the results in the paper showing two different odors, I feel that it is a very tricky claim that the intertwined pair of trajectories indicates homogeneity more than the other pair does. More experiments can improve the results and I believe that another round of reviewing is encouraged to improve the quality. ------------------------------------ Originality: The paper is well motivated and organized, and of great significance to brain functional connectivity research. However, there are some clarity issues and technical details that might need improvement. Here is a list of my concerns: 1. On pg 1, “The model 21 produces strong statistical evidence for DFC…” I am confused about what is the specific statistical evidence. Since the experimental results in the paper haven’t provided significant statistical evidence, I would suggest the authors’ statement be more concrete. 2. On pg 2, “\Lambda a p\times r matrix of factor loadings” should be “r \times p”. 3. On pg 3, in eq (5), p_2 and p_3 are not defined. 4. On pg 4, it would be better to explain more insight of Th 1. What does weakly stationary property imply except “The covariance of the log-covariance process Y (t) depends only on the static loading matrix and the factor covariance kernels”. 5. The authors claim that “the model posterior allows us to perform Bayesian inference for scientific hypotheses, for instance, whether the LFP time series is non-stationary, and if characteristics of DFC differ across experimental conditions”. It is not very clear to me how the authors justify these statements via their experimental results presented in the paper. 6. I would expect the authors illustrate more details of the process, by which they chose the specific 6 channels. Fig 5 show correlation between 2 of the 6 channels. Could the authors tell how they choose the specific 2 out of the 6? 7. Fig. 5 shows two median GP factors visualized as trajectories in latent space can be separated based on the odor. Since there is a lack of analysis on the statistical significance, it would be highly recommended to replicate the experiments on other odors to validate the separation between two distinct odors and similarity between two similar odors. 8. I would expect that the author shows that the problem of factorizing Y_i(t) into F_i(t)*B is well-defined. It means that F_i(t) could be in a so over-complicated space that the factorization result of B and F_i(t) becomes very unstable and sensitive to the initialization.

Reviewer 3



After rebuttal -------------------- Thank you for the rebuttal. Most of my concerns are addressed in the rebuttal to some degree. However, I am still missing a couple of details, e.g. - better justification of the sliding window "likelihood" - a discussion of how the proposed inference scheme affects the exact posterior when they add factors sequentially rather than fitting all factors jointly - I still think that the LPF experiment would be much stronger if the authors had included a baseline method for comparison. For these reasons, my score of 6 remains unchanged. Summary -------------------------- The paper proposes a new model for analyzing dynamic functional connectivity (DFC). The goal is to estimate the dynamics of the second order moments from a set of non-stationary multidimensional time-series. The authors propose to first estimate a sequence of time-varying covariance matrices using a “sliding window”-estimator and then model these estimates using a factor model with Gaussian process priors on the factors. Rather than modelling the sliding window estimates directly, they model the matrix logarithm of these estimates using a factor model to ensure that the resulting covariance matrix is positive semi-definite at all times. The author further state three properties of the model: weak stationarity, the posterior contraction rate, and large prior support. They propose a sampling-based method for inference. The paper is concluded with two numerical experiment: a quantitative experiment based on a simulated data and a qualitative experiment based on real data (local field potentials from rats) but without any baseline comparison. Clarity ------------------- Overall, the paper is well organized, well-written, and easy to follow. However, some aspects of the proposed inference algorithm is unclear and should be improved (see comments below) . Quality ------------------------- The paper appears technically sound. The authors provided the source code for the proposed method. Originality & Significance ------------------------------------------------ The paper proposes a novel way to analyze dynamic functional connectivity, which is an active area of research in neuroscience, and thus, the paper is likely to be of interest to the neuroscience community. However, the idea of parametrizing covariance matrices using the matrix logarithm map in general is not new. Furthermore, the authors provide an interesting analysis of local field potential data from rats. However, there are two issues that makes it difficult to asses the significance of the contribution: I) Some experimental details are unclear. For example, in lines 235-238, the authors state that the analyze data is for one particular rat out of five and they focus on 78 trials out of 200 trials. The authors should clarify how the subset of data is chosen, i.e. are the specific rat and the subset of trials chosen randomly before analysing the data? II) The authors do not provide any baseline results for experiment. The experiment would be more convincing if the authors included a qualitative or quantitative baseline. Other comments -------------------------------- The authors should elaborate on the motivation for modelling the sliding window estimates rather than modelling the data directly. That is, \mathcal{D}(0, K_i(t)) in eq. (2) could be replaced with a proper likelihood. In the past, several papers have shown that the estimating dynamic functional connectivity using sliding window estimators can be problematic [1, 2, 3]. Therefore, the authors should discuss if the concerns about the sliding window estimators also apply to their method. How is the window size L and the taper parameter tau chosen? Can they be inferred from the data? The paper [4] proposes a different Gaussian proposed-based model for functional connectivity. This work seems highly relevant, so the author should discuss and compare to two models. The details of the inference scheme unclear. In section 3.3 the author states that they use the Hamiltonian Monte Carlo with No-U-turn sampler to sample the posterior distribution of the loadings and in Section 3.4 they describe how they fit the model using a Gibbs sampler. The author should explain the interplay between these algorithms more carefully. They could consider including an algorithm with pseudo-code in the supplementary material. In line 162 the authors state that “... fit the new factor on the residuals”. Does this mean that all previous factors are kept fixed when adding new factors? If so, the authors should clarify how this affects the posterior distribution The authors mention that they use 5000 MCMC (with 1000 warm-up) samples. However, sampling from GP and horseshoe priors can be difficult, so it would be beneficial to compute and state the standard convergence diagnostics for the MCMC chains, e.g. Rhat and effective sample sizes etc. Minor comments -------------------------------- Line 83: The dimensions of Lambda seems to be flipped. Shouldn’t it be r x p rather than p x r? Line 111: The authors state that “\bm{u} maps a matrix to its vectorized upper triangle”. This seems flipped as well because K = Log(u(Y)), where Y is a q-vector. Line 123: The following statement is unclear to me “The model posteriors are conditioned on difference observations despite sharing the same kernel” Line 131: The author should probably state explicitly that (B)_kj = \beta_{kj} References ----------------------- [1] Laumann et al: On the stability of bold fmri correlations. Cerebral Cortex, 2016. [2] Lindquist et al: Evaluating dynamic bivariate correlations in resting-state fmri: A comparison study and a new approach. Neuroimage, 2014 [3] Hindriks et al: Can sliding-window correlations reveal dynamic functional connectivity in resting-state fMRI? Neuroimage, 2016. [4] Andersen et al: Bayesian Structure Learning for Dynamic Brain Connectivity, AISTATS, 2017

[Author Response · NeurIPS 2019]

We would like to thank the reviewers for taking the time to provide us with helpful feedback and will definitely incorporate the suggestions in the final version. Below are our clarifications for the questions raised.

**Weak Stationarity** Since weak stationarity is a key assumption for many theorems in time series analysis, we felt it was important to explicitly show this property. In particular, weak stationarity implies that the model curves lie in a Hilbert space, permitting direct applications of Bochner's theorem, the Fourier transform, and other spectral methods. Further, the proof of weak stationarity provides an explicit formula for the covariance of $Y$, which produces the covariance structure given in Figure 2(d). In practice, the assumption of a weakly stationary covariance process can be empirically checked to determine whether the LFGP model is appropriate.

**Factor modeling** In the present work, the factors have a GP prior distribution with squared exponential kernel, which is a distribution over the functions in the RKHS defined by this kernel. With this kernel, the factorization is equivalent to a multivariate time series factor model (Lopes & West) with an AR(1) covariance, which has been extensively studied and validated. The stability of the factorization under other kernels will be important to study in future work.

**Sliding window** As our inference goals were solely regarding the covariance process, we chose to forgo specifying an explicit distribution on the observed time series. Since the LFGP model is fit across repeated trials, the artifacts introduced by sliding window estimation are unlikely to influence the modeling results, assuming that the artifacts are uncorrelated across trials. In order to conduct inference on the theta band oscillations in the DFC series, we determined an L between 50-200ms to be scientifically reasonable. L was set to 100ms to reduce some noise without oversmoothing and 15ms taper was chosen, which gives a tapered window with similar structure to that used in Lindquist 2014.

**LFP data** The local field potentials are measured by surgically implanted tetrodes and the exact tetrode locations vary across rats. Therefore, it may not make sense to compare LFP channels of different rats. This issue actually motivates the latent factor approach because we want to eventually visualize and compare the latent trajectories for all the rats. We have focused on the data from a particular rat because it has the best memory task performance so the data has the strongest signal. The six LFP channels are chosen also to help with the signal to noise ratio because they have the most neurons attached. Lastly, we used trials of odor B and C because the rats smell the first odor (A) much more often than odors in the end (D, E) due to the experimental design.

**Model fitting** In Section 3.3, the horseshoe prior is used on the loadings as an illustration of the methodology when the number of factors is unknown. A short chain of the No-U-Turn Sampler is run within each iteration of the Gibbs sampler. For analyzing the LFP data, we actually used the Gaussian-Inverse Gamma conjugate priors for the loadings and variances. As for the $\mathcal{GP}$ hyper-parameters, we used much more informative priors because of concerns about convergence and identifiability. Gamma(5, 1) prior is used for the length scales because five time points span 100ms and it still has diffused mass from 0 to 10. Based on the trace plots, we can see good convergence and sufficient shift from the prior. The effective sample size is 276 (median) for the loadings but falls short for the length scales.

**Interpretation of results** For non-stationarity and odor separation, we did not perform any statistical test because Bayesian hypothesis testing in general and the null hypothesis formulation in this case are not well-defined. However, we would argue that there are many possible ways to interpret the results. From the posterior correlation between two selected channels, we can observe that it changes from negative to positive so the covariance is varying to some extent. As for odor separation, when we fit the model to two random subsets of the 58 trials of odor A, the latent trajectories are much more intertwined. We can repeat similar analysis for more odors and for resting data.

**Model comparison** Comparing dynamic connectivity models whose approaches range from classical time series to graph based models is tricky; every paper has a different simulation study but at least the "truth" is known. On the LFP data, we know there could be dynamic connectivity and odor differentiation, but not for a certainty. Therefore, it is difficult to find a single "benchmark model" that estimates dynamic connectivity and differentiates odors. That said, the model in Andersen et al is similar to our model in the way it utilizes latent Gaussian processes. It would be very interesting to compare the two models on a publicly available data set with published results as suggested.

[Meta-Review · NeurIPS 2019]

The manuscript proposes a probabilistic model for dynamic functional connectivity. Novel components include enforcing positive definiteness using a log representation. The reviewers discuss the novelty of the modeling approach and the theoretical claims as strengths of the manuscript. On the other hand, the reviewers point out major weaknesses which are primarily empirical, including missing details on key hyper-parameters, missing comparison to standard baselines, and insufficient empirical validation i.e. using more than one real dataset. Overall, the reviewers and AC agree that this is a timely and interesting submission, which can be improved in terms of empirical evaluation.